# Weighted QMIX: Expanding Monotonic Value Function Factorisation for Deep Multi-Agent Reinforcement Learning

**Tabish Rashid, Gregory Farquhar**[*]**, Bei Peng, Shimon Whiteson**
Department of Computer Science
University of Oxford
{tabish.rashid, gregory.farquhar, bei.peng, shimon.whiteson}@cs.ox.ac.uk

## Abstract

QMIX is a popular $Q$-learning algorithm for cooperative MARL in the centralised training and decentralised execution paradigm. In order to enable easy decentralisation, QMIX restricts the joint action $Q$-values it can represent to be a monotonic *mixing* of each agent's utilities. However, this restriction prevents it from representing value functions in which an agent's ordering over its actions can depend on other agents' actions. To analyse this representational limitation, we first formalise the objective QMIX optimises, which allows us to view QMIX as an operator that first computes the $Q$-learning targets and then projects them into the space representable by QMIX. This projection returns a representable $Q$-value that minimises the unweighted squared error across all joint actions. We show in particular that this projection can fail to recover the optimal policy even with access to $Q^*$, which primarily stems from the equal weighting placed on each joint action. We rectify this by introducing a weighting into the projection, in order to place more importance on the better joint actions. We propose two weighting schemes and prove that they recover the correct maximal action for any joint action $Q$-values, and therefore for $Q^*$ as well. Based on our analysis and results in the tabular setting, we introduce two scalable versions of our algorithm, Centrally-Weighted (CW) QMIX and Optimistically-Weighted (OW) QMIX and demonstrate improved performance on both predator-prey and challenging multi-agent StarCraft benchmark tasks [26].

## 1 Introduction

Many critical tasks involve multiple agents acting in the same environment. To learn good behaviours in such problems from agents' experiences, we may turn to multi-agent reinforcement learning (MARL). Fully decentralised policies are often used in MARL, due to practical communication constraints or as a way to deal with an intractably large joint action space. However, when training in simulation or under controlled conditions we may have access to additional information, and agents can freely share their observations and internal states. Exploiting these possibilities can greatly improve the efficiency of learning [7, 9].

In this paradigm of centralised training for decentralised execution, QMIX [25] is a popular $Q$-learning algorithm with state-of-the-art performance on the StarCraft Multi-Agent Challenge [26]. QMIX represents the optimal joint action value function using a monotonic *mixing* function of per-agent utilities. This restricted function class $Q^{mix}$ allows for efficient maximisation during training, and easy decentralisation of the learned policy. However, QMIX is unable to represent joint action value functions that are characterised as *nonmonotonic* [16], i.e., an agent's ordering over its

---

[*]Now at Google DeepMind.

own actions depends on other agents' actions. Consequently, QMIX cannot solve tasks that require significant coordination within a given timestep [27, 2]. In this work, we analyse an idealised, tabular version of QMIX to study this representational limitation, and then develop algorithms to resolve these limitations in theory and in practice.

We formalise the objective that QMIX optimises, which allows us to view QMIX as an operator that first computes the $Q$-learning targets and then projects them into $\mathcal{Q}^{mix}$ by minimising the unweighted squared error across all joint actions. We show that, since in general $Q^* \notin \mathcal{Q}^{mix}$, the projection of $Q^*$, which we refer to as $Q_{tot}$, can have incorrect estimates for the optimal joint action, yielding suboptimal policies. These are fundamental limitations of the QMIX algorithm independent from exploration and compute constraints, and occur even with access to the true $Q^*$.

These limitations primarily arise because QMIX's projection of $Q^*$ yields a $Q_{tot}$ that places equal importance on approximating the $Q$-values for *all* joint actions. Our key insight is that if we ultimately care only about the greedy optimal policy, it is more important to accurately represent the value of the optimal joint action than the suboptimal ones. Therefore, we can improve the policy recovered from $Q_{tot}$ by appropriately weighting each joint action when projecting $Q^*$ into $\mathcal{Q}^{mix}$.

Based on this intuition, we introduce a weighting function into our projection. In the idealised tabular setting we propose two weighting functions and prove that the projected $Q_{tot}$ recovers the correct maximal action for any $Q$, and therefore for $Q^*$ as well. Since this projection always recovers the correct maximal joint action, we benefit from access to $Q^*$ (or a learned approximation of it). To this end, we introduce a learned approximation of $Q^*$, from an unrestricted function class, which we call $\hat{Q}^*$. By using $Q_{tot}$, now a weighted projection of $\hat{Q}^*$, to perform maximisation, we show that $\hat{Q}^*$ converges to $Q^*$ and that $Q_{tot}$ thus recovers the optimal policy.

Based on our analysis and results in the tabular setting, we present two scalable versions of our algorithm, Centrally-Weighted (CW) QMIX and Optimistically-Weighted (OW) QMIX. We demonstrate their improved ability to cope with environments with nonmonotonic value functions, by showing superior performance in a predator-prey task in which the trade-offs made by QMIX prevent it from solving the task. Additionally, we demonstrate improved robustness over QMIX to the amount of exploration performed, by showing better empirical performance on a range of SMAC maps. Our ablations and additional analysis experiments demonstrate the importance of both a weighting and an unrestricted $\hat{Q}^*$ in our algorithm.

## 2 Background

A fully cooperative multi-agent sequential decision-making task can be described as a *decentralised partially observable Markov decision process* (Dec-POMDP) [21] consisting of a tuple $G = \langle S, U, P, r, Z, O, n, \gamma \rangle$. $s \in S$ describes the true state of the environment. At each time step, each agent $a \in A \equiv \{1, ..., n\}$ chooses an action $u_a \in U$, forming a joint action $\mathbf{u} \in \mathbf{U} \equiv U^n$. This causes a transition on the environment according to the state transition function $P(s'|s, \mathbf{u}) : S \times \mathbf{U} \times S \to [0, 1]$. All agents share the same reward function $r(s, \mathbf{u}) : S \times \mathbf{U} \to \mathbb{R}$ and $\gamma \in [0, 1)$ is a discount factor.

Due to the *partial observability*, each agent's individual observations $z \in Z$ are produced by the observation function $O(s, a) : S \times A \to Z$. Each agent has an action-observation history $\tau_a \in T \equiv (Z \times U)^*$, on which it conditions a (potentially stochastic) policy $\pi^a(u_a|\tau_a) : T \times U \to [0, 1]$. $\boldsymbol{\tau}$ denotes the action-observation histories of all agents (up to that timestep). The joint policy $\pi$ has a joint *action-value function*: $Q^\pi(s_t, \mathbf{u}_t) = \mathbb{E}_{s_{t+1:\infty}, \mathbf{u}_{t+1:\infty}}[R_t|s_t, \mathbf{u}_t]$, where $R_t = \sum_{i=0}^{\infty} \gamma^i r_{t+i}$ is the *discounted return*. For our idealised tabular setting, we consider a *fully observable* setting in which each agent's observations are the full state. This is equivalent to a multi-agent MDP (MMDP) [21] which is itself equivalent to a standard MDP with $U^n$ as the action space.

We adopt the *centralised training and decentralised execution* paradigm [20, 14]. During training our learning algorithm has access to the true state $s$ and every agent's action-observation history, as well as the freedom to share all information between agents. However, during testing (execution), each agent has access only to its own action-observation history.

## 2.1 QMIX

The Bellman optimality operator is defined by:

$$\mathcal{T}^* Q(s, \mathbf{u}) := \mathbb{E}[r + \gamma \max_{\mathbf{u}'} Q(s', \mathbf{u}')], \tag{1}$$

where the expectation is over the next state $s' \sim P(\cdot|s, \mathbf{u})$ and reward $r \sim r(\cdot|s, \mathbf{u})$. Algorithms based on $Q$-learning [34] use samples from the environment to estimate the expectation in (1), in order to update their estimates of $Q^*$.

VDN [29] and QMIX are $Q$-learning algorithms for the cooperative MARL setting, which estimate the optimal joint action value function $Q^*$ as $Q_{tot}$, with specific forms. VDN factorises $Q_{tot}$ into a sum of the per-agent utilities: $Q_{tot}(s, \mathbf{u}) = \sum_{a=1}^{n} Q_a(s, u_a)$, whereas QMIX combines the per-agent utilities via a continuous monotonic function that is state-dependent: $f_s(Q_1(s, u_1), ..., Q_n(s, u_n)) = Q_{tot}(s, \mathbf{u})$, where $\frac{\partial f_s}{\partial Q_a} \geq 0,\ \forall a \in A \equiv \{1, ..., n\}$.

In the deep RL setting neural networks with parameters $\theta$ are used as function approximators, and QMIX is trained much like a DQN [19]. Considering only the fully-observable setting for ease of presentation, a replay buffer stores transition tuples $(s, \mathbf{u}, r, s', d)$, in which the agents take joint action $\mathbf{u}$ in state $s$, receive reward $r$ and transition to $s'$ and $d$ is a boolean indicating if $s'$ is a terminal state. QMIX is trained to minimise the squared TD error on a minibatch of $b$ samples from the replay buffer: $\sum_{i=1}^{b}(Q_{tot}(s, \mathbf{u}; \theta) - y_i)^2$, where $y_i = r + \gamma \max_{\mathbf{u}'} Q_{tot}(s', \mathbf{u}'; \theta^-)$ are the targets, and $\theta^-$ are the parameters of a *target network* that are periodically copied from $\theta$. The monotonic *mixing* function $f_s$ is parametrised as a feedforward network, whose non-negative weights are generated by hypernetworks [10] that take the state as input.

## 3 QMIX Operator

In this section we examine an operator that represents an *idealised* version of QMIX in a tabular setting. The purpose of our analysis is primarily to understand the fundamental limitations of QMIX that stem from its training objective and the restricted function class it uses. We write this function class as $\mathcal{Q}^{mix}$ :

$$\mathcal{Q}^{mix} := \{Q_{tot}|Q_{tot}(s, \mathbf{u}) = f_s(Q_1(s, u_1), ...Q_n(s, u_n)), \frac{\partial f_s}{\partial Q_a} \geq 0, Q_a(s, u) \in \mathbb{R}\}.$$

This is the space of all $Q_{tot}$ that can be represented by monotonic funtions of tabular $Q_a(s, u)$. At each iteration of our idealised algorithm, we constrain $Q_{tot}$ to lie in $\mathcal{Q}^{mix}$ by solving the following optimisation problem:

$$\underset{q \in \mathcal{Q}^{mix}}{\mathrm{argmin}} \sum_{\mathbf{u} \in \mathbf{U}} (\mathcal{T}^* Q_{tot}(s, \mathbf{u}) - q(s, \mathbf{u}))^2. \tag{2}$$

To avoid any confounding factors regarding exploration, we assume this optimisation is performed for all states and joint actions at each iteration, as in planning algorithms like value iteration [24]. We also assume the optimisation is performed exactly. However, since it is not guaranteed to have a unique solution, a random $q$ is returned from the set of objective-minimising candidates.

By contrast, the original QMIX algorithm in the deep RL setting interleaves exploration and approximate optimisation of this objective, using samples from the environment to approximate $\mathcal{T}^*$, and a finite number of gradient descent steps to estimate the argmin. Additionally, sampling uniformly from a replay buffer of the most recent experiences does not strictly lead to a uniform weighting across joint actions. Instead the weighting is proportional to the frequency at which a joint action was taken. Incorporating this into our analysis would introduce significant complexity and detract from our main focus: to analyse the limitations of restricting the representable function class to $\mathcal{Q}^{mix}$.

The optimisation in (2) can be separated into two distinct parts: the first computes targets using $\mathcal{T}^*$, and the second projects those targets into $\mathcal{Q}^{mix}$. We define the corresponding projection operator $\Pi_{\mathrm{Qmix}}$ as follows:

$$\Pi_{\mathrm{Qmix}} Q := \underset{q \in \mathcal{Q}^{mix}}{\mathrm{argmin}} \sum_{\mathbf{u} \in \mathbf{U}} (Q(s, \mathbf{u}) - q(s, \mathbf{u}))^2$$

We can then define $\mathcal{T}^*_{\mathrm{Qmix}} := \Pi_{\mathrm{Qmix}} \mathcal{T}^*$ as the QMIX operator, which exactly corresponds to the objective in (2).

There is a significant literature studying projections of value functions into linear function spaces for RL (see [30] for a detailed introduction and overview). However, despite some superficial similarities, our focus is considerably different in several ways. First, we consider a nonlinear projection, rendering many methods of analysis inapplicable. Second, as the $Q_a(s, u)$ are tabular, there is no tradeoff in the quality of representation across different states, and consequently no need to weight the different states in the projection. By contrast, unlike linear $Q$-functions, our restricted space does induce tradeoffs in the quality of represenation across different joint actions, and weighting them differently in the projection is central to our method. Third, the targets in our optimisation are fixed to $\mathcal{T}^* Q_{tot}(s, \mathbf{u})$ at each iteration, rather than depending on $q$ as they would in a minimisation of Mean Squared Bellman Error (MSBE) or Mean Squared Projected Bellman Error (MSPBE) in linear RL. This makes our setting closer to that of fitted $Q$-iteration [6] in which a regression problem is solved at each iteration to fit a function approximator from a restricted class to $Q$-learning-style targets. Our focus of study is the unique properties of the particular function space $\mathcal{Q}^{mix}$, and the tradeoffs in representation quality induced by projection into it.

### 3.1 Properties of $\mathcal{T}^*_{\mathbf{Qmix}}$

To highlight the pitfalls of the projection $\Pi_{\text{Qmix}}$ into $\mathcal{Q}^{mix}$, we consider the effect of applying $\Pi_{\text{Qmix}}$ to the true $Q^*$, which is readily available in deterministic normal form games where $Q^*$ is just the immediate reward.

$\mathcal{T}^*_{\mathbf{Qmix}}$ **is not a contraction.** The payoff matrix in Table 1 (Left) is a simple example of a value function that cannot be perfectly represented in $\mathcal{Q}^{mix}$. Table 1 (Middle) and (Right) show two distinct $Q_{tot}$, both of which are global minima of the optimisation solved by $\Pi_{\text{Qmix}} Q^*$. Hence, $\mathcal{T}^*_{\text{Qmix}}$ is not a contraction, which would have a unique fixed point.

| 1 | 0 |
|---|---|
| 0 | 1 |

| 1 | 1/3 |
|---|---|
| 1/3 | 1/3 |

| 1/3 | 1/3 |
|---|---|
| 1/3 | 1 |

Table 1: Non-monotonic payoff matrix (Left) and the two possible $Q_{tot}$'s returned by $\Pi_{\text{Qmix}}$ (Middle and Right).

**QMIX's argmax is not always correct.** There exist $Q$-functions such that $\text{argmax}\,\Pi_{\text{Qmix}} Q \neq \text{argmax}\,Q$. For example, the payoff matrix in Table 2 (Left) (from Son et al. [27]) produces a value function for which QMIX's approximation (Right) does not result in the correct argmax.

**QMIX can underestimate the value of the optimal joint action.** Furthermore, if it has an incorrect argmax, the value of the true optimal joint action can be underestimated, e.g., $-12$ instead of 8 in Table 2. If QMIX gets the correct argmax then it represents the maximum $Q$-value perfectly (proved formally in Appendix B). However, if QMIX's argmax joint action is not the true optimal joint action then QMIX can underestimate the value of that action.

| 8 | -12 | -12 |
|---|---|---|
| -12 | 0 | 0 |
| -12 | 0 | 0 |

| -12 | -12 | -12 |
|---|---|---|
| -12 | 0 | 0 |
| -12 | 0 | 0 |

Table 2: Payoff matrix (Left) in which $Q_{tot}$ returned from $\Pi_{\text{Qmix}}$ has an incorrect argmax (Right).

These failure modes are problematic because they show fundamental limitations of QMIX, that are independent from: 1) compute constraints, since we exactly minimise the objectives posed; 2) exploration, since we update every state-action pair; and 3) parametrisation of the mixing function and agent utilities, since we are assume that $Q_{tot}$ can be any member of $\mathcal{Q}^{mix}$, whereas in practice we can only represent a subset of $\mathcal{Q}^{mix}$.

## 4 Weighted QMIX Operator

In this section we introduce an operator for an *idealised* version of our algorithm, *Weighted QMIX* (WQMIX), in order to compare it to the operator we introduced for QMIX.

The negative results in Section 3.1 concern the scenario in which we optimise QMIX's loss function across *all* joint actions for every state. We argue that this equal weighting over joint actions when performing the optimisation in (2) is responsible for the possibly incorrect argmax of the objective-minimising solution. Consider the example in Table 2. A monotonic $Q_{tot} \in \mathcal{Q}^{mix}$ cannot increase its estimate of the value of the single optimal joint action above -12 without either increasing the estimates of the value of the bad joint actions above their true value of -12, or decreasing the estimates of the zero-value joint actions below -12. The error for misestimating several of the suboptimal joint

actions would outweigh the improvement from better estimating the single optimal joint action. As a result the optimal action value is underestimated and the resulting policy is suboptimal.

By contrast, consider the extreme case in which we only optimise the loss for the *single* optimal joint action $\mathbf{u}^*$. For a single action, the representational limitation of QMIX has no effect so we can optimise the objective perfectly, recovering the value of the optimal joint action.

However, we still need to learn that the other action values are lower than $Q_{tot}(\mathbf{u}^*)$ in order to recover the optimal policy. To prioritise estimating $Q_{tot}(\mathbf{u}^*)$ well, while still anchoring down the value estimates for other joint actions, we can add a suitable weighting function $w$ into the projection operator of QMIX:

$$\Pi_w Q := \underset{q \in \mathcal{Q}^{mix}}{\operatorname{argmin}} \sum_{\mathbf{u} \in \mathbf{U}} w(s, \mathbf{u})(Q(s, \mathbf{u}) - q(s, \mathbf{u}))^2. \tag{3}$$

The weighting function $w : S \times \mathbf{U} \to (0, 1]$ weights the importance of each joint action in QMIX's loss function. It can depend on more than just the state and joint action, but we omit this from the notation for simplicity. Setting $w(s, \mathbf{u}) = 1$ recovers the projection operator $\Pi_{\text{Qmix}}$.

## 4.1 Weightings

The choice of weighting is crucial to ensure that WQMIX can overcome the limitations of QMIX. As shown in Section 3, even if we have access to $Q^*$, if we use a uniform weighting then we can still end up with the wrong argmax after projection into the monotonic function space. We now consider two different weightings and show in Theorems 1 and 2 that these choices of $w$ ensure that the $Q_{tot}$ returned from the projection has the correct argmax. The proofs of these theorems can be found in Appendix B. For both weightings, let $\alpha \in (0, 1]$ and consider the weighted projection of an arbitrary joint action value function $Q$.

**Idealised Central Weighting**    The first weighting, which we call Idealised Central Weighting, is quite simple:

$$w(s, \mathbf{u}) = \begin{cases} 1 & \mathbf{u} = \mathbf{u}^* = \operatorname{argmax}_{\mathbf{u}} Q(s, \mathbf{u}) \\ \alpha & \text{otherwise.} \end{cases} \tag{4}$$

To ensure that the weighted projection returns a $Q_{tot}$ with the correct argmax, we simply down-weight every suboptimal action. However, this weighting requires computing the maximum across the joint action space, which is often infeasible. In Section 5 we discuss an approximation to this weighting in the deep RL setting.

**Theorem 1.** *Let $w$ be the Idealised Central Weighting from* (4). *Then $\exists \alpha > 0$ such that* $\operatorname{argmax} \Pi_w Q = \operatorname{argmax} Q$ *for any $Q$.*

Theorem 1 provides a sanity check that this choice of weighting guarantees we recover a $Q_{tot}$ with the correct argmax in this idealised setting, with a nonzero weighting for suboptimal actions.

**Optimistic Weighting**    The second weighting, which we call Optimistic Weighting, affords a practical implementation:

$$w(s, \mathbf{u}) = \begin{cases} 1 & Q_{tot}(s, \mathbf{u}) < Q(s, \mathbf{u}) \\ \alpha & \text{otherwise.} \end{cases} \tag{5}$$

This weighting assigns a higher weighting to those joint actions that are underestimated relative to $Q$, and hence could be the true optimal actions (in an optimistic outlook).

**Theorem 2.** *Let $w$ be the Optimistic Weighting from* (5). *Then $\exists \alpha > 0$ such that,* $\operatorname{argmax} \Pi_w Q = \operatorname{argmax} Q$ *for any $Q$.*

Theorem 2 shows that Optimistic Weighting also recovers a $Q_{tot}$ with the correct argmax.

## 4.2 Weighted QMIX Operators

We have shown that these two weightings are guaranteed to recover the correct maximum joint action for any $Q$, and therefore for $Q^*$ as well. This is in contrast to the uniform weighting ($w = 1$) of

QMIX, which can fail to recover the correct optimal joint action even for simple matrix games. The weighted projection now allows us to fully take advantage of $Q^*$.

Since we do not have access to the true optimal value function in general, we learn an approximation to it: $\hat{Q}^*$, which does not need to lie in the restricted monotonic function space $\mathcal{Q}^{mix}$. Performing an exact maximisation of $\hat{Q}^*$ requires a search over the entire joint action space, which is typically intractable and does not admit decentralisation. We instead use our QMIX approximation $Q_{tot}$ to *suggest* the maximum joint action(s), which can then be evaluated by $\hat{Q}^*$.

Learning $\hat{Q}^*$ instead of using $Q_{tot}$ in its place brings some advantages. First, it allows us a richer representational class to approximate $Q^*$ with, since we place no restrictions on the form of $\hat{Q}^*$. In the idealised tabular setting, $Q^*$ is exactly representable by $\hat{Q}^*$. Second, since we are weighting each joint action in $\Pi_w$, $Q_{tot}$ (unlike $\hat{Q}^*$) likely has less accurate estimates for those joint actions with a low weighting. Due to these factors, we may bootstrap using more accurate estimates by using $\hat{Q}^*$ instead of $Q_{tot}$. These properties are necessary to ensure that WQMIX converges to the optimal policy. The operator used to update $\hat{Q}^*$ is:

$$\mathcal{T}_w^* \hat{Q}^*(s, \mathbf{u}) := \mathbb{E}[r + \gamma \hat{Q}^*(s', \underset{\mathbf{u}'}{\operatorname{argmax}} Q_{tot}(s', \mathbf{u}'))]. \tag{6}$$

Since $Q_{tot}$ is monotonic, the argmax in (6) is tractable. Similarly $Q_{tot}$ is updated in tandem using:

$$\mathcal{T}_{\text{WQMIX}}^* Q_{tot} := \Pi_w \mathcal{T}_w^* \hat{Q}^* \tag{7}$$

$\mathcal{T}_w^*$ is similar to the Bellman Optimality Operator in (1) but does not directly maximise over $\hat{Q}^*$. Instead it uses $Q_{tot} \in \mathcal{Q}^{mix}$ to suggest a maximum joint action. Setting $w$ to be uniform ($w = 1$) here does not recover QMIX since we are additionally learning $\hat{Q}^*$.

Finally, using our previous results, we show that $\hat{Q}^*$ converges to $Q^*$ and that $Q_{tot}$ recovers an optimal policy. This provides a firm theoretical foundation for Weighted QMIX: in an idealised setting it converges to the optimal policy, whereas QMIX does not.

**Corollary 1.** *Letting $w$ be the Idealised Central or Optimistic Weighting, then $\exists \alpha > 0$ such that the unique fixed point of $\mathcal{T}_w^*$ is $Q^*$. Furthermore, $\Pi_w Q^* \subseteq \mathcal{Q}^{mix}$ recovers an optimal policy, and $\max \Pi_w Q^*(s, \cdot) = \max Q^*(s, \cdot)$.*

In this section we have shown that an idealised version of Weighted QMIX can converge to $Q^*$ and recover an optimal policy. Restricting $Q_{tot}$ to lie in $\mathcal{Q}^{mix}$ does not prevent us from representing an optimal policy, since there is *always* an optimal deterministic policy [24] and all deterministic policies can be derived from the argmax of a $Q$ that lies in $\mathcal{Q}^{mix}$. Thus, we do *not* expand the function class that we consider for $Q_{tot}$. Instead, we change the solution of the projection by introducing a weighting.

## 5  Deep RL Algorithm

So far, we have only considered an idealised setting in order to analyse the fundamental properties of QMIX and Weighted QMIX. However, the ultimate goal of our analysis is to inform the development of new scalable RL algorithms, in combination with, e.g., neural network function approximators. We now describe the realisation of Weighted QMIX for deep RL, in a Dec-POMDP setting in which each agent does not observe the full state, as described in Section 2.

There are three components to Weighted QMIX: 1) $Q_{tot}$, i.e., the per-agent utilities $Q_a$ (from which the decentralised policies are derived) and the mixing network, 2) an unrestricted joint action $\hat{Q}^*$, and 3) a weighting function $w$, as in $\Pi_w$.

$\mathbf{Q_{tot}}$  The $Q_{tot}$ component is largely the same as that of Rashid et al. [25], using the architecture from Samvelyan et al. [26]. $Q_{tot}$ is trained to minimise the following loss:

$$\sum_{i=1}^{b} w(s, \mathbf{u})(Q_{tot}(\boldsymbol{\tau}, \mathbf{u}, s) - y_i)^2, \tag{8}$$

where $y_i := r + \gamma \hat{Q}^*(s', \boldsymbol{\tau}', \operatorname{argmax}_{\mathbf{u}'} Q_{tot}(\boldsymbol{\tau}', \mathbf{u}', s'))$ is treated as a fixed target. This differs from the idealised setting considered in Sections 3 and 4 because we are now only optimising the $Q$-values

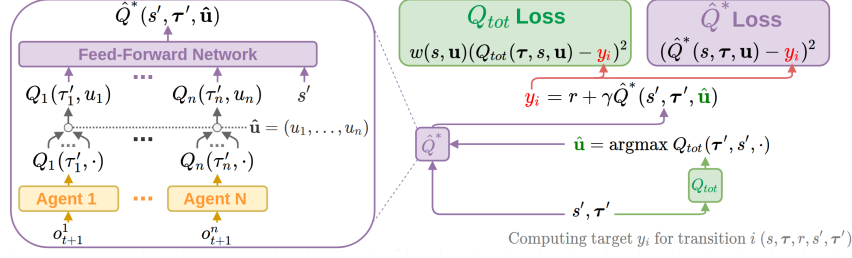

Figure 1: Deep RL Weighted QMIX setup. **Left:** The architecture used for $\hat{Q}^*$. **Right:** How the targets $y_i$ for each transition in the minibatch are computed and used.

for the state-action pairs present in the minibatch sampled from the replay buffer, as opposed to every state-action pair.

**Centralised $\hat{Q}^*$** We use a similar architecture to $Q_{tot}$ to represent $\hat{Q}^*$: agent networks whose chosen action's utility is fed into a mixing network. $\hat{Q}^*$ is thus conditioned on the state $s$ and the agents' action-observation histories $\boldsymbol{\tau}$. For the agent networks we use the same architecture as QMIX. They do not share parameters with the agents used in representing $Q_{tot}$. The mixing network for $\hat{Q}^*$ is a feed-forward network that takes the state and the appropriate actions' utilities as input. This mixing network is not constrained to be monotonic by using non-negative weights. Consequently, we can simplify the architecture by having the state and agent utilities be inputs to $\hat{Q}^*$'s mixing network, as opposed to having hypernetworks take the state as input and generate the weights. The architecture of $\hat{Q}^*$ is shown in Figure 1 (Left). $\hat{Q}^*$ is trained to minimise the following loss, using $y_i$ from (8):

$$\sum_{i=1}^{b}(\hat{Q}^*(s, \boldsymbol{\tau}, \mathbf{u}) - y_i)^2. \tag{9}$$

**Weighting Function** Idealised Central Weighting requires knowing the maximal joint action over $\hat{Q}^*$, which is computationally infeasible. In order to derive a practical algorithm, we must make approximations. For each state-action pair in the sampled minibatch, the weighting we use is:

$$w(s, \mathbf{u}) = \begin{cases} 1 & y_i > \hat{Q}^*(s, \boldsymbol{\tau}, \hat{\mathbf{u}}^*) \ \text{or} \ \mathbf{u} = \hat{\mathbf{u}}^* \\ \alpha & \text{otherwise,} \end{cases} \tag{10}$$

where $\hat{\mathbf{u}}^* = \text{argmax}_{\mathbf{u}} Q_{tot}(\boldsymbol{\tau}, \mathbf{u}, s)$. Since we do not know the maximal joint action for each state, we make a local approximation: if $y_i > \hat{Q}^*(s, \boldsymbol{\tau}, \hat{\mathbf{u}}^*)$, then $\mathbf{u}$ might be the best joint action. We use $\hat{Q}^*(s, \boldsymbol{\tau}, \hat{\mathbf{u}}^*)$ instead of $\mathcal{T}_w^* \hat{Q}^*(s, \boldsymbol{\tau}, \hat{\mathbf{u}}^*)$ since we do not have direct access to it. We refer to this weighting function as **Centrally-Weighted QMIX (CW)**.

The Optimistic Weighting presented in (5) does not require any approximations. The exact weighting we use is:

$$w(s, \mathbf{u}) = \begin{cases} 1 & Q_{tot}(\boldsymbol{\tau}, \mathbf{u}, s) < y_i \\ \alpha & \text{otherwise.} \end{cases}$$

We refer to it as **Optimistically-Weighted QMIX (OW)**.

In a deep RL setting, QMIX implicitly weights the joint-actions proportional to their execution by the behaviour policies used to fill the replay buffer. This forces QMIX to make trade-offs in its $Q$-value approximation that are directly tied to the exploration strategy chosen. However, as we have shown earlier, this can lead to poor estimates for the optimal joint action and thus yield suboptimal policies. Instead, Weighted QMIX separates the weighting of the joint actions from the behaviour policy. This allows us to focus our monotonic approximation of $Q^*$ on the *important* joint actions, thus encouraging better policies to be recovered irrespective of the exploration performed.

# 6 Results

In this section we present our experimental results on the Predator Prey task considered by Bohmer et al. [2] and on a variety of SMAC[2] scenarios. More details about the implementation of each are included in Appendix C, as well as additional ablation experiments (Appendix E). For every graph we plot the median and shade the 25%-75% quartile. Code is available at `https://github.com/oxwhirl/wqmix`.

## 6.1 Predator Prey

We first consider a partially-observable Predator Prey task involving 8 agents [2], which was designed to test coordination between agents by providing a punishment of -2 reward when only a single agent (as opposed to two agents) attempt to capture a prey. Algorithms which suffer from *relative overgeneralisation* [23, 35], or which make poor trade-offs in their representation (as VDN and QMIX do) can fail to solve this task.

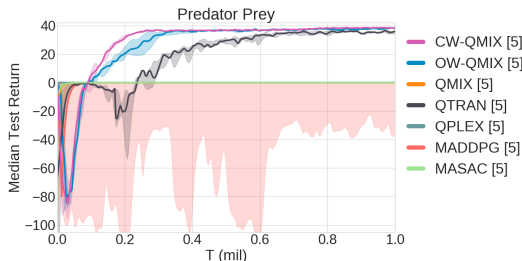

Figure 2: Median test return for the predator prey environment, comparing Weighted QMIX and baselines.

As shown in [2], QMIX fails to learn a policy that achieves positive test reward, and our results additionally show the same negative results for MADDPG and MASAC. Interestingly, QPLEX also fails to solve the task despite not having any restrictions on the joint-action $Q$-values it can represent, suggesting difficulties in learning certain value functions. Figure 2 shows that both CW-QMIX and OW-QMIX solve the task faster than QTRAN.

## 6.2 SMAC

### 6.2.1 Robustness to increased exploration

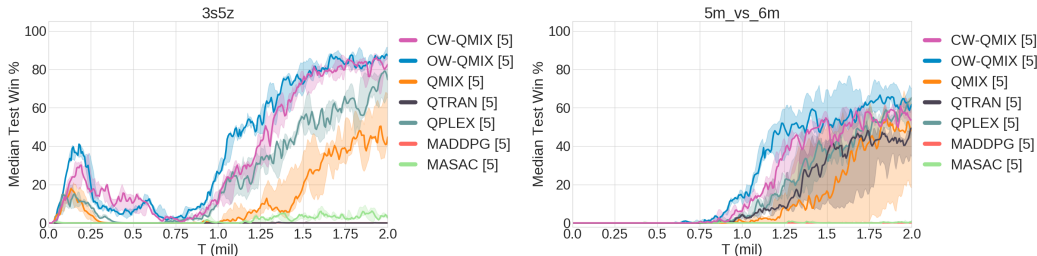

Figure 3: Median test win % with an increased rate of exploration.

QMIX is particularly brittle when there is significant exploration being done, since it then tries (and fails) to represent the value of many suboptimal joint actions. We would like our algorithms to efficiently learn from exploratory behaviour. Hence, we use two of the easier SMAC maps, *3s5z* and *5m_vs_6m*, to test the robustness of our algorithms to exploration. We use an $\epsilon$-greedy policy in which $\epsilon$ is annealed from 1 to 0.05 over 1 million timesteps, increased from the $50k$ used in [26]. Figure 3 shows the results of these experiments, in which both Weighted QMIX variants significantly outperform all baselines. Figure 4 (Left) additionally compares the performance of QMIX and Weighted QMIX on `bane_vs_bane`, a task with 24 agents, across two $\epsilon$-schedules. We can see that both variants of Weighted QMIX are able to solve the task irrespective of the level of exploration, whereas QMIX fails to do so.

### 6.2.2 Necessity of increased exploration

Next, we compare our method on the challenging *6h_vs_8z*, which is classified as a *super hard* SMAC map due to current method's poor performance [26]. Figure 4 (Right) compares QMIX and Weighted

[2]We utilise `SC2.4.6.2.69232` (the same version as [26]) instead of the newer `SC2.4.10`. Performance is **not** comparable across versions.

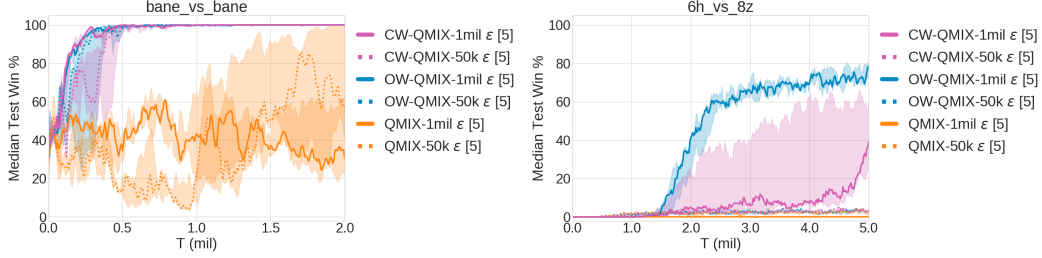

Figure 4: Median test win % with 2 exploration schedules on `bane_vs_bane` and *6h_vs_8z*.

QMIX with two differing exploration schedules (annealing $\epsilon$ over 50k or 1 million timesteps, denoted -50k $\epsilon$ and -1mil $\epsilon$ respectively). We can see that a larger rate of exploration is required, and only Weighted QMIX can successfully recover a good policy, demonstrating the benefits of our method for improving performance in a challenging coordination problem.

### 6.2.3 Limitations

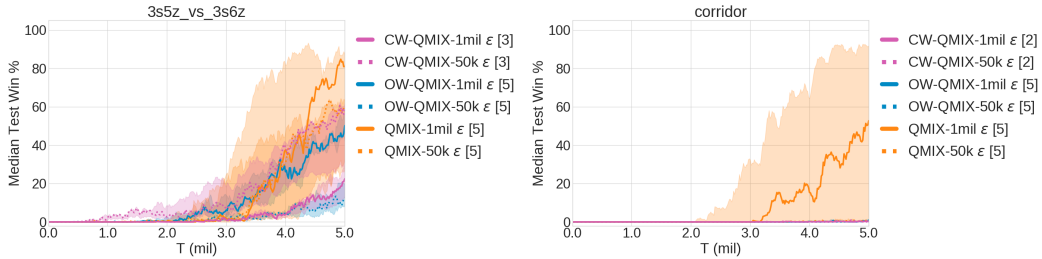

Figure 5: Median test win % with 2 exploration schedules on 2 *super-hard* SMAC maps.

Finally, we present results on 2 more *super hard* SMAC maps in order to show the limitations of our method in Figure 5. In *3s5z_vs_3s6z* we observe that the extra exploration is not helpful for any method. Since QMIX is almost able to solve the task, this indicates that both exploration and the restricted function class of QMIX are not limiting factors in this scenario. On *corridor* we see that only QMIX with an extended exploration schedule manages non-zero performance, showing the importance of sufficient exploration on this map. The poor performance of Weighted QMIX shows that the extra complexity of our method (notably learning $\hat{Q}^*$) can sometimes harm performance, indicating that closer attention must be paid to the architecture and weighting functions.

Figure 6 shows the results on *corridor* for Weighted QMIX with a slightly modified architecture for $\hat{Q}^*$. The significantly improved performance for Weighted QMIX indicates that the architecture of $\hat{Q}^*$ is partly responsible for the regression in performance over QMIX.

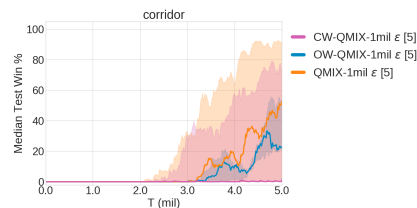

Figure 6: Median test win % on *corridor* with a modified architecture for $\hat{Q}^*$.

## 7 Conclusions and Future Work

This paper presented Weighted QMIX, which was inspired by analysing an idealised version of QMIX that first computes the $Q$-learning targets and then projects them into $\mathcal{Q}^{mix}$. QMIX uses an unweighted projection that places the same emphasis on every joint action, which can lead to suboptimal policies. Weighted QMIX rectifies this by using a weighted projection that allows more emphasis to be placed on *better* joint actions. We formally proved that for two specific weightings, the weighted projection is guaranteed to recover the correct maximal joint action for any $Q$. To fully take advantage of this, we additionally learn an unrestricted joint action $\hat{Q}^*$, and prove that it converges to $Q^*$. We extended Weighted QMIX to deep RL and showed its improved ability to coordinate and its robustness to an increased rate of exploration. For future work, more complicated weightings could be considered, as opposed to the simplicitic weightings we used where $w$ is either 1 or $\alpha$ in this paper. Additionally, some of our results demonstrate the limitations of our method, partially stemming from the architecture used for $\hat{Q}^*$.

## Broader Impact

Due to the broad applicability of cooperative Multi-Agent Reinforcement Learning, we limit our discussion to the cooperative setting in which agents must act independently without communication. One such potential application is in self-driving cars, in which the agents should be able to make safe and sensible decisions even without access to a communication network. Due to the sample inefficiency of current RL methods, combined with their lack of safe exploration it is also necessary to first train agents in a simulated setting before even beginning to train in the real world. Hence, the class of algorithms we consider in this paper could be used to train agents in these scenarios and would likely be chosen over fully-decentralised options. It is important then to obtain a better understanding of the current approaches, in particular of their limitations. In this paper, we focus primarily on the limitations of QMIX due to its strong performance [26]. We also investigate the links between QTRAN and our algorithm and observe poor empirical performance for Actor-Critic style approaches. Investigating all of these further should improve the performance of all algorithms in this domain, and provide a better understanding of their relative strengths and weaknesses.

One particular limitation of QMIX is that it can fail in environments in which an agent's best action is dependent on the actions the other agents take, i.e., in environments in which agents must *coordinate* at the same timestep. However, in a Multi-Agent setting it is often *crucial* to coordinate with the other agents. Our approach lifts this restriction, and is theoretically able to learn the optimal policy in any environment which greatly increases its applicability. Extending the capabilities of cooperative MARL algorithms should further extend the applicability of these algorithms in a broader range of applications. However, our approach introduces extra complexity and can perform poorly in certain challenging domains. It is important then to consider whether the extra modelling capacity of our method is required to achieve good performance on a selected task.

## Acknowledgments and Disclosure of Funding

We thank the members of the Whiteson Research Lab for their helpful feedback. This project has received funding from the European Research Council (ERC), under the European Union's Horizon 2020 research and innovation programme (grant agreement number 637713). It was also supported by an EPSRC grant (EP/M508111/1, EP/N509711/1) and the UK EPSRC CDT in Autonomous Intelligent Machines and Systems. The experiments were made possible by a generous equipment grant from NVIDIA. Shimon Whiteson is the Head of Research at Waymo, UK.

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
