[Supplementary Material]

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

# A Related Work

In this section we briefly describe related work on cooperative MARL in the common paradigm of Centralised Training and Decentralised Execution (CTDE). For a more comprehensive survey of cooperative MARL, see [22].

Tampuu et al. [31] train each agent's DQN [19] using Independent $Q$-learning [32], which treats the other agents as part of the environment, which can lead to many pathologies and instability during training [4, 8]. By contrast VDN [29] and QMIX [25] learn the joint-action $Q$-values, which avoids some of these issues. Qatten [37] change the architecture of QMIX's mixing network to a 2-layer linear mixing, in which the weights of the first layer are produced through an attention-based mechanism. Bohmer et al. [3] learn a centralised joint action $Q$-Value that is approximately maximised by coordinate descent and used to generate trajectories that IQL agents train on. SMIX($\lambda$) [38] replaces the 1-step $Q$-learning target with a SARSA($\lambda$) target. Despite claiming their method can represent a larger class of joint action $Q$-values, they can represent exactly the same class as QMIX ($Q^{mix}$) since they use the same architecture (in particular, non-negative weights in the mixing network). Yang et al. [36] utilise integrated gradients to decompose the joint-action $Q$-values of the critic into individual utilities for each agent, a form of multi-agent credit assignment. The agents are then regressed against their respective utilities.

Mahajan et al. [16] point out some limitations of QMIX arising from its monotonic factorisation. Specifically, they show that for a specific choice of matrix game, QMIX can fail to learn an optimal policy if each joint action is visited uniformly, which corresponds to our idealised tabular setting in Section 3. Additionally, they show that a lower bound on the probability of recovering the optimal policy increases for an $\epsilon$-greedy policy as $\epsilon$ increases. This is proved by considering the weighting on each joint-action induced by the exploration policy. By contrast, our weighting is independent of the exploration strategy, adding flexibility. Optimistic Weighting uses a smaller weighting when decreasing $Q_{tot}$ estimates, similarly to Hysteretic $Q$-learning [17] which uses a smaller learning rate when decreasing value estimates of independent learners.

**Relationship to Actor-Critic.** Weighted QMIX bears many similarities to an off-policy actor-critic algorithm, if we view $\hat{Q}^*$ as the critic and the policy implied by $Q_{tot}$ as the actor. Define the deterministic QMIX greedy policy (assuming full observability to simplify the presentation) as:

$$\pi_w(s) = \begin{pmatrix} \text{argmax}_{u_1} Q_1(s, u_1) \\ \dots \\ \text{argmax}_{u_n} Q_n(s, u_n) \end{pmatrix}.$$

Weighted QMIX trains $\hat{Q}^*$ to approximate $Q^{\pi_w}$, the $Q$-values of this policy. This is also an approximation to $Q$-learning since $\pi_w \approx \text{argmax} \hat{Q}^*$. Viewed in this manner, Weighted QMIX is similar to MADDPG [15] with a single critic, except for how the actors are trained. MADDPG trains each agent's policy $\pi_a$ via the multi-agent deterministic policy gradient theorem , whereas Weighted QMIX trains the policy indirectly by training $Q_{tot}$ via the weighted loss in (8). Multi-agent Soft Actor Critic (MASAC) [11, 12] is another off-policy actor-critic based approach that instead trains the actors by minimising the KL divergence between each agent's policies and the joint-action $Q$-values. These actor-critic based approaches (as well as COMA [9] and LIIR [5]) do not restrict the class of joint action $Q$-values they can represent, which theoretically allows them to learn an optimal value function and policy. However, in practice they do not perform as well as QMIX, perhaps due to relative overgeneralisation [35] or the presence of bad local minima.

**Relationship To QTRAN.** QTRAN [27] is another $Q$-learning based algorithm that learns an unrestricted joint action $Q$ function and aims to solve a constrained optimisation problem in order to decentralise it. However, it is empirically hard to scale to more complex tasks (such as SMAC).

We can view QTRAN as specific choices of the 3 components of Weighted QMIX, which allows us to better understand its trade-offs and empirical performance in relation to WQMIX. However, the motivations for QTRAN are significantly different. $\mathbf{Q_{tot}}$ is represented using VDN instead of QMIX, and trained using $\hat{Q}^*$ as the target (instead of $y_i$). This can limit QTRAN's empirical performance because QMIX generally outperforms VDN [26]. $\hat{\mathbf{Q}}^*$ is a network that takes as input an embedding of all agents' chosen actions and observations (and additionally the state if it is available). The agent components share parameters with the agent networks used to approximate $Q_{tot}$.

**Weighting.** The weighting function is as follows:

$$w(s, \mathbf{u}) = \begin{cases} \lambda_{opt} & \mathbf{u} = \hat{\mathbf{u}} \\ \lambda_{nopt} & Q_{tot}(s, \mathbf{u}) < \hat{Q}^*(s, \mathbf{u}) \\ 0 & \text{otherwise,} \end{cases} \tag{11}$$

where $\hat{\mathbf{u}} = \operatorname{argmax} Q_{tot}(s, \cdot)$. Using too small a weight in the weighting can have a substantial negative effect on performance (as we show in Appendix E). However, using a 0 weight for overestimated $Q$-values is a fundamental part of the QTRAN algorithm.

**Concurrent Work.** Wang et al. [33] propose QPLEX, which also expands the class of joint-action $Q$-values that can be represented. They acheive this by decomposing $Q_{tot}$ as a sum of a value function and a non-positive advantage function. Crucially, this advantage function is 0 for the joint-action in which every agent maximises their own utilities ($\hat{\mathbf{u}}^*$ in our notation). This ensures consistency between the agent's greedy joint-action ($\hat{\mathbf{u}}^*$), and the true maximum joint-action of $Q_{tot}$. In contrast, Weighted QMIX does not maintain this consistency. Our experimental results show that despite not restricting the class of joint-action $Q$-values that can be represented, QPLEX can struggle to learn a good policy in some environments like QMIX.

Son et al. [28] propose QOPT, which also learns an unrestricted $\hat{Q}^*$ and utilises an optimistic-style weighting to train $Q_{tot}$ that is represented by QMIX. In contrast to Weighted QMIX, whose $\hat{Q}^*$ does not share any parameters with $Q_{tot}$, the unrestricted joint-action $Q$-values of QOPT are obtained through an unrestricted mixing network (the weights are not constrained to be non-negative) which takes as input the agent utilities. The weighting function used for training $Q_{tot}$ is similar to the weighting for CW-QMIX, in which a smaller weighting is used for the the joint-actions whose $Q$-values are estimated as lower than the $Q$-values for the approximate best joint-action ($\hat{\mathbf{u}}^*$).

# B  Proof of Theorems

**Proposition 1.** *For any* $w : S \times \mathbf{U} \to (0, 1]$ *and* $Q$. *Let* $Q_{tot} = \Pi_w Q$. *Then* $\forall s \in S$, $\hat{\mathbf{u}} \in \operatorname{argmax} Q_{tot}(s, \cdot)$. *We have that* $Q_{tot}(s, \hat{\mathbf{u}}) \geq Q(s, \hat{\mathbf{u}})$. *If* $\hat{\mathbf{u}} = \mathbf{u}^* := \operatorname{argmax}_{\mathbf{u}} Q(s, \mathbf{u})$ *then* $Q_{tot}(s, \hat{\mathbf{u}}) = Q(s, \hat{\mathbf{u}})$.

*Proof.* Consider a $s \in S$. Assume for a contradiction that $Q_{tot}(s, \hat{\mathbf{u}}) < Q(s, \hat{\mathbf{u}})$.

Define $Q'_{tot}$ as follows:

$$Q'_{tot}(s, \mathbf{u}) = \begin{cases} Q(s, \mathbf{u}) & \mathbf{u} = \hat{\mathbf{u}} \\ Q_{tot}(s, \mathbf{u}) & \text{otherwise,} \end{cases}$$

By construction we have that $Q'_{tot} \in \mathcal{Q}^{mix}$, and

$$\begin{aligned} &\sum_{\mathbf{u} \in \mathbf{U}} w(s, \mathbf{u})(Q(s, \mathbf{u}) - Q'_{tot}(s, \mathbf{u}))^2 \\ &= \sum_{\mathbf{u} \neq \hat{\mathbf{u}}} w(s, \mathbf{u})(Q(s, \mathbf{u}) - Q'_{tot}(s, \mathbf{u}))^2 + w(s, \hat{\mathbf{u}})(Q(s, \hat{\mathbf{u}}) - Q'_{tot}(s, \hat{\mathbf{u}}))^2 \\ &= \sum_{\mathbf{u} \neq \hat{\mathbf{u}}} w(s, \mathbf{u})(Q(s, \mathbf{u}) - Q'_{tot}(s, \mathbf{u}))^2 \\ &\qquad (Q'_{tot}(s, \hat{\mathbf{u}}) = Q(s, \hat{\mathbf{u}})) \\ &= \sum_{\mathbf{u} \neq \hat{\mathbf{u}}} w(s, \mathbf{u})(Q(s, \mathbf{u}) - Q_{tot}(s, \mathbf{u}))^2 \\ &\qquad (Q'_{tot}(s, \mathbf{u}) = Q_{tot}(s, \mathbf{u}) \; \forall \mathbf{u} \neq \hat{\mathbf{u}}) \\ &< \sum_{\mathbf{u} \neq \hat{\mathbf{u}}} w(s, \mathbf{u})(Q(s, \mathbf{u}) - Q_{tot}(s, \mathbf{u}))^2 + w(s, \hat{\mathbf{u}})(Q(s, \hat{\mathbf{u}}) - Q_{tot}(s, \hat{\mathbf{u}}))^2 \\ &\qquad (Q_{tot}(s, \hat{\mathbf{u}}) < Q(s, \hat{\mathbf{u}})) \\ &= \sum_{\mathbf{u} \in \mathbf{U}} w(s, \mathbf{u})(Q(s, \mathbf{u}) - Q_{tot}(s, \mathbf{u}))^2. \end{aligned}$$

Thus $Q_{tot}$ cannot be the solution of $\Pi_w Q$, a contradiction. And so $Q_{tot}(s, \hat{\mathbf{u}}) \geq Q(s, \hat{\mathbf{u}})$.

Now consider the scenario in which $\hat{\mathbf{u}} = \mathbf{u}^*$, and assume for a contradiction that $Q_{tot}(s, \hat{\mathbf{u}}) > Q(s, \hat{\mathbf{u}})$.

Define $Q'_{tot}$ as follows:

$$Q'_{tot}(s, \mathbf{u}) = \begin{cases} Q(s, \mathbf{u}) & \mathbf{u} = \hat{\mathbf{u}} = \mathbf{u}^* \\ \min\{Q_{tot}(s, \mathbf{u}), Q(s, \mathbf{u}^*)\} & \text{otherwise,} \end{cases}$$

Again, by construction $Q'_{tot} \in \mathcal{Q}^{mix}$.

$$\sum_{\mathbf{u} \in \mathbf{U}} w(s, \mathbf{u})(Q(s, \mathbf{u}) - Q'_{tot}(s, \mathbf{u}))^2$$

$$= \sum_{\mathbf{u} \neq \hat{\mathbf{u}}} w(s, \mathbf{u})(Q(s, \mathbf{u}) - Q'_{tot}(s, \mathbf{u}))^2 + w(s, \hat{\mathbf{u}})(Q(s, \hat{\mathbf{u}}) - Q'_{tot}(s, \hat{\mathbf{u}}))^2$$

$$= \sum_{\mathbf{u} \neq \hat{\mathbf{u}}} w(s, \mathbf{u})(Q(s, \mathbf{u}) - Q'_{tot}(s, \mathbf{u}))^2$$

$$\quad (Q'_{tot}(s, \hat{\mathbf{u}}) = Q(s, \hat{\mathbf{u}}))$$

$$\leq \sum_{\mathbf{u} \neq \hat{\mathbf{u}}} w(s, \mathbf{u})(Q(s, \mathbf{u}) - Q_{tot}(s, \mathbf{u}))^2$$

$$\quad (\text{If } \min\{Q_{tot}(s, \mathbf{u}), Q(s, \hat{\mathbf{u}})\} = Q_{tot}(s, \mathbf{u}) \text{ then } Q'_{tot}(s, \mathbf{u}) = Q_{tot}(s, \mathbf{u}).$$

$$\quad \text{Otherwise } Q(s, \mathbf{u}^*) < Q_{tot}(s, \mathbf{u}) \implies (Q(s, \mathbf{u}^*) - Q(s, \mathbf{u}))^2 < (Q_{tot}(s, \mathbf{u}) - Q(s, \mathbf{u}))^2,$$

$$\quad \text{since } Q(s, \mathbf{u}) \leq Q(s, \mathbf{u}^*).)$$

$$< \sum_{\mathbf{u} \neq \hat{\mathbf{u}}} w(s, \mathbf{u})(Q(s, \mathbf{u}) - Q_{tot}(s, \mathbf{u}))^2 + w(s, \hat{\mathbf{u}})(Q(s, \hat{\mathbf{u}}) - Q_{tot}(s, \hat{\mathbf{u}}))^2$$

$$\quad (Q_{tot}(s, \hat{\mathbf{u}}) > Q(s, \hat{\mathbf{u}}))$$

$$= \sum_{\mathbf{u} \in \mathbf{U}} w(s, \mathbf{u})(Q(s, \mathbf{u}) - Q_{tot}(s, \mathbf{u}))^2.$$

Thus, $Q_{tot}$ cannot be the solution of $\Pi_w Q$, a contradiction. This proves that $Q_{tot}(s, \hat{\mathbf{u}}) = Q(s, \hat{\mathbf{u}})$ if $\hat{\mathbf{u}} = \mathbf{u}^*$. □

**Proposition 2.** *Let $Q_{tot} = \Pi_w Q$. $\forall s \in S \; \exists \hat{\mathbf{u}} \in \arg\max Q_{tot}(s, \cdot)$ such that $Q_{tot}(s, \hat{\mathbf{u}}) = Q(s, \hat{\mathbf{u}})$.*

*Proof.* Assume for a contradiction that $\forall \hat{\mathbf{u}} \in \arg\max Q_{tot}$ we have that $Q_{tot}(\hat{\mathbf{u}}) > Q(\hat{\mathbf{u}})$.

Define $\Delta_s := Q_{tot}(s, \hat{\mathbf{u}}) - \max\{Q_{tot}(s, \mathbf{u}) | \mathbf{u} \in \mathbf{U}, Q_{tot}(s, \mathbf{u}) < Q_{tot}(s, \hat{\mathbf{u}})\}$ to be the difference between the maximum $Q$-Value and the next biggest $Q$-Value (the action gap [1]). $\Delta_s$ is well defined as long as there exists a sub-optimal action. If there is not a suboptimal action, then trivially any $\mathbf{u} \in \mathbf{U}$ satisfies the condition.

Let $\epsilon = \min\{\Delta_s/2, (Q_{tot}(s, \hat{\mathbf{u}}) - \max\{Q(s, \mathbf{u}) | \mathbf{u} \in \arg\max Q_{tot}(s, \cdot)\})/2\} > 0$.

Define $Q'_{tot}$ as follows:

$$Q'_{tot}(s, \mathbf{u}) = \begin{cases} Q_{tot}(s, \mathbf{u}) - \epsilon & \mathbf{u} \in \arg\max Q_{tot} \\ Q_{tot}(s, \mathbf{u}) & \text{otherwise.} \end{cases}$$

i.e. we have decreased the $Q$-Value estimates for the argmax joint actions by a small non-zero amount. Since $\epsilon < \Delta_s$ we do not need to worry about adjusting other action's estimates.

By construction $Q'_{tot} \in \mathcal{Q}^{mix}$.

Then $Q'_{tot}$ has a smaller loss than $Q_{tot}$ since the estimates for the argmax actions are closer to the true values.

This gives our contradiction since $Q_{tot} \in \Pi_w Q$.

Thus $\exists \hat{\mathbf{u}} \in \arg\max Q_{tot}$ such that $Q_{tot}(\hat{\mathbf{u}}) \leq Q(\hat{\mathbf{u}})$. Combined with Proposition 1 gives us the required result. $\square$

**Corollary 2.** *If $Q_{tot}$ has a unique argmax $\hat{\mathbf{u}}$, then $Q_{tot}(\hat{\mathbf{u}}) = Q(\hat{\mathbf{u}})$.*

*Proof.* Proposition 2 showed the existence of an argmax action whose $Q_{tot}$-value matches $Q$ exactly. If there is a unique argmax $\hat{\mathbf{u}}$, then it must match exactly giving our result. $\square$

**Theorem 1.** *Let $w$ be the Idealised Central Weighting from Equation* (4). *Then $\exists \alpha > 0$ such that,* $\arg\max \Pi_w Q = \arg\max Q$ *for any $Q$.*

**Theorem 2.** *Let $w$ be the Optimistic Weighting from Equation* (5). *Then $\exists \alpha > 0$ such that,* $\arg\max \Pi_w Q = \arg\max Q$ *for any $Q$.*

*Proof.* Since the proof of both theorems contains a significant overlap, we will merge them both into a single proof.

We will start by first considering the Idealised Central Weighting: Let $Q_{tot} = \Pi_w Q$ ($Q_{tot} \in \Pi_w Q$ if there are distinct solutions).

Let $\mathbf{u}^* \in \arg\max Q$, be an optimal action.

Consider a state $s \in S$.

Define $\Delta_s := Q(s, \mathbf{u}^*) - \max\{Q(s, \mathbf{u})|\mathbf{u} \in \mathbf{U}, Q(s, \mathbf{u}) < Q(s, \mathbf{u}^*)\}$ to be the difference between the maximum $Q$-Value and the next biggest $Q$-Value (the action gap [1]). $\Delta_s$ is well defined as long as there exists a sub-optimal action. If there is not a suboptimal action, then trivially $\arg\max \Pi_w Q = \arg\max Q$ for state $s$.

Let $\hat{\mathbf{u}} \in \arg\max \Pi_w Q$, and consider the loss when $\hat{\mathbf{u}} = \mathbf{u}^*$.

By Propositon 1 we have that $Q_{tot}(s, \hat{\mathbf{u}}) = Q(s, \hat{\mathbf{u}})$.

Then the loss:

$$\sum_{\mathbf{u} \in \mathbf{U}} w(s, \mathbf{u})(Q(s, \mathbf{u}) - Q_{tot}(s, \mathbf{u}))^2 = \alpha \sum_{\mathbf{u} \neq \mathbf{u}^*} (Q(s, \mathbf{u}) - Q_{tot}(s, \mathbf{u}))^2 < \alpha(\frac{R_{max}}{1-\gamma})^2 |U|^n,$$

where $R_{max} := \max r - \min r$. The last inequaility follows since the maximum difference between $Q$-values in the discounted setting is then $\frac{R_{max}}{1-\gamma}$, and there are $|U|^n$ joint-actions total.

Whereas if $\hat{\mathbf{u}} \neq \mathbf{u}^*$, then the loss

$$\sum_{\mathbf{u} \in \mathbf{U}} w(s, \mathbf{u})(Q(s, \mathbf{u}) - Q_{tot}(s, \mathbf{u}))^2$$
$$= (Q(s, \mathbf{u}^*) - Q_{tot}(s, \mathbf{u}^*))^2 + \alpha \sum_{\mathbf{u} \neq \mathbf{u}^*} (Q(s, \mathbf{u}) - Q_{tot}(s, \mathbf{u}))^2$$
$$\geq \Delta_s^2,$$

since $Q(s, \mathbf{u}^*) - Q_{tot}(s, \mathbf{u}^*) \geq \Delta_s$, which is proved below.

By Proposition 2 let $\hat{\mathbf{u}}' \in \arg\max Q_{tot}$ such that $Q_{tot}(s, \hat{\mathbf{u}}') = Q(s, \hat{\mathbf{u}}')$. Then $Q(s, \mathbf{u}^*) \geq \Delta_s + Q(s, \hat{\mathbf{u}}') = \Delta_s + Q_{tot}(s, \hat{\mathbf{u}}') > \Delta_s + Q_{tot}(s, \mathbf{u}^*) \implies Q(s, \mathbf{u}^*) - Q_{tot}(s, \mathbf{u}^*) > \Delta_s$. The strict inequaility $Q_{tot}(s, \hat{\mathbf{u}}') > Q_{tot}(s, \mathbf{u}^*)$ used is due to $\hat{\mathbf{u}} \neq \mathbf{u}^*$.

Setting $0 < \alpha_s < \frac{\Delta_s^2 (1-\gamma)^2}{(R_{max})^2 |U|^n}$ then gives the required result for state $s$.

Letting $\alpha = \min_s \alpha_s > 0$ completes the proof for the Idealised Central Weighting.

For the proof of the Optimistic Weighting we will use many of the same arguments and notation.

We will once again consider a single state $s \in S$ and the action gap $\Delta_s$.

Now, let us consider a $Q_{tot}$ of a specific form: For $s \in S$, let $Q_{tot}(s, \hat{\mathbf{u}}) = c_s + \epsilon$, where $\epsilon << \Delta_s$ and $Q_{tot}(s, \mathbf{u}) = c_s, \forall \mathbf{u} \neq \hat{\mathbf{u}}$. Note that here $Q_{tot}$ has a unique maximum action.

For this $Q_{tot}$, consider $\hat{\mathbf{u}} = \mathbf{u}^*$ then the loss:

$$\sum_{\mathbf{u} \in \mathbf{U}} w(s, \mathbf{u})(Q(s, \mathbf{u}) - Q_{tot}(s, \mathbf{u}))^2 = \alpha \sum_{\mathbf{u} \neq \mathbf{u}^*} (Q(s, \mathbf{u}) - Q_{tot}(s, \mathbf{u}))^2 \leq \alpha(\frac{R_{max}}{1-\gamma})^2 |U|^n < \Delta_s^2,$$

since $Q_{tot}(s, \mathbf{u}) = c_s = Q(s, \mathbf{u}^*) - \epsilon > Q(s, \mathbf{u}), \forall \mathbf{u} \neq \mathbf{u}^*$ by Proposition 1, which means that $w(s, \mathbf{u} \neq \mathbf{u}^*) = \alpha$. The final inequality follows due to setting $0 < \alpha_s < \frac{\Delta_s^2(1-\gamma)^2}{(R_{max})^2|U|^n}$ as earlier.

Now consider any $Q'_{tot} \in \mathcal{Q}^{mix}$.

If for this $Q'_{tot}$, $\hat{\mathbf{u}} \neq \mathbf{u}^*$, then $w(s, \mathbf{u}^*) = 1$ and thus the loss

$$\sum_{\mathbf{u} \in \mathbf{U}} w(s, \mathbf{u})(Q(s, \mathbf{u}) - Q_{tot}(s, \mathbf{u}))^2$$
$$= (Q(s, \mathbf{u}^*) - Q_{tot}(s, \mathbf{u}^*))^2 + \sum_{\mathbf{u} \neq \mathbf{u}^*} w(s, \mathbf{u})(Q(s, \mathbf{u}) - Q_{tot}(s, \mathbf{u}))^2$$
$$\geq \Delta_s^2.$$

By Proposition 2 let $\hat{\mathbf{u}}' \in \operatorname{argmax} Q'_{tot}$ such that $Q'_{tot}(s, \hat{\mathbf{u}}') = Q(s, \hat{\mathbf{u}}')$. Since $Q(s, \mathbf{u}^*) \geq \Delta_s + Q(s, \hat{\mathbf{u}}') = \Delta_s + Q_{tot}(s, \hat{\mathbf{u}}') > \Delta_s + Q_{tot}(s, \mathbf{u}^*) \implies Q(s, \mathbf{u}^*) - Q_{tot}(s, \mathbf{u}^*) > \Delta_s$.

Thus, we have shown that for any $Q'_{tot}$ with $\hat{\mathbf{u}} \neq \mathbf{u}^*$ the loss is greater than the $Q_{tot}$ we first considered with $\hat{\mathbf{u}} = \mathbf{u}^*$.

And so for state $s$, $\operatorname{argmax} \Pi_w Q(s, \cdot) = \mathbf{u}^* = \operatorname{argmax} Q(s, \cdot)$.

Letting $\alpha = \min_s \alpha_s > 0$ once again completes the proof. $\qquad \square$

**Corollary 3.** *Letting $w$ be the Central or Optimistic Weighting, then $\exists \alpha > 0$ such that the unique fixed point of $\mathcal{T}_w^*$ is $Q^*$. Furthermore, $\Pi_w Q^* \subseteq \mathcal{Q}^{mix}$ recovers an optimal policy, and $\max \Pi_w Q^* = \max Q^*$.*

*Proof.* Using the results of Theorems 1 and 2 we know that $\exists \alpha > 0$ such that $\operatorname{argmax} \Pi_w Q = \operatorname{argmax} Q$. We also know from their proofs that the same $\alpha$ works for both weightings.

Instead of updating $Q_{tot}$ in tandem with $\hat{Q}^*$, we can instead write $\mathcal{T}_w^*$ as:

$$\mathcal{T}_w^* \hat{Q}^*(s, \mathbf{u}) := \mathbb{E}[r + \gamma \hat{Q}^*(s', \operatorname{argmax}_{\mathbf{u}'}(\Pi_w \hat{Q}^*)(s', \mathbf{u}'))].$$

And so:

$$\hat{Q}^*(s', \operatorname{argmax}_{\mathbf{u}'}(\Pi_w \hat{Q}^*)(s', \mathbf{u}')) = \max_{\mathbf{u}'} \hat{Q}^*(s', \mathbf{u}'), \; \forall s' \in S.$$

Thus, our operator $\mathcal{T}_w^*$ is equivalent to the usual Bellman Optimality Operator $\mathcal{T}^*$, which is known to have a unique fixed point $Q^*$ [34, 18].

Once again by the results of Theorems 1 and 2, we know that $Q_{tot}^* \in \Pi_w Q^*$ acheives the correct argmax for every state. Thus it is an optimal policy. Finally, Proposition 1 shows that $\max \Pi_w Q^* = \max Q^*$.

$\qquad \square$

## C   Experimental Setup

We adopt the same training setup as [26], using PyMARL to run all experiments. The architecture for QMIX is also the same as in [26].

The architecture of the mixing network for $\hat{Q}^*$is a feed forward network with 3 hidden layers of 256 dim and ReLU non-linearities. Shown in Figure 1.

For the experiment in Figure 6 the architecture for $\hat{Q}^*$is modified slightly. We replace the first hidden layer with a hypernetwork layer. A hypernetwork with a single hidden layer of dim 64 (with

ReLU) takes the state as input and generates the weight matrix. Inspired by [37] we then take the column-wise softmax of this weight matrix, which can be viewed as an approximation of multi-head attention.

The architecture and setup for QTRAN is also the same, except we use a 3 layer feedforward network of dim $\{64, 256\}$ to match the depth of $\hat{Q}^*$.

MADDPG and MASAC's critic shares the same architecture as $\hat{Q}^*$.

MADDPG is trained using the deterministic multi-agent policy-gradient theorem, via the Gumbel-Softmax trick, as in [15, 12]. Specifically for agent $a$ we produce $Q(s, u_a, u^{-a})$ (with target network $Q$) for each possible action, where $u^{-a}$ are the actions of the other agents produced by their most recent policies. We multiply these by the agent's policy (one-hot vector since it is deterministic) and use the Straight Through Gumbel-Softmax estimator [13] to differentiate through this with a temperature of 1.

MASAC is trained by minimising the KL divergence between each agent's policy $\pi_a$ and $\exp(Q(s, \cdot, u^{-a}) - \alpha_{ent} \log \pi_a)$., Since the KL divergence is an expectation: $\mathbb{E}_{\pi_a}[\log(\frac{\pi_a}{\exp(Q(s, \cdot, u^{-a}) - \alpha_{ent} \log \pi_a)})]$, we approximate it by sampling an action from $\pi_a$ for each agent. These sampled actions are used for $u^{-a}$. For the actor's policies we use the same $\epsilon$-greedy floor technique as in [9].

QPLEX uses the same setup for its mixing network as for the SMAC experiments in [33].

## C.1 Predator Prey

For Weighted QMIX variants (and ablations with just a weighting), we consider $\alpha \in \{0.1, 0.5\}$ and set $\alpha = 0.1$ for all variants.

For QTRAN we set $\lambda_{opt} = 1$ and consider $\lambda_{nopt} \in \{0.1, 1, 10\}$ (since only their relative weighting makes a difference), and the dim of the mixing network in $\{64, 256\}$. We set $\lambda_{nopt} = 10$ and the dim of the mixing network to 64.

For MASAC we consider $\alpha_{ent} \in \{0, 0.001, 0.01, 0.1\}$ and set it to 0.001.

## C.2 SMAC Robustness to exploration

For Weighted QMIX we consider $\alpha \in \{0.01, 0.1, 0.5, 0.75\}$ and set $\alpha = 0.75$ for CW and $\alpha = 0.5$ for OW. All lines are available in Appendix E.

For the Weighted QMIX ablations we considered $\alpha \in \{0.5, 0.75\}$ and set $\alpha = 0.75$.

For QTRAN we set $\lambda_{opt} = 1$ and consider $\lambda_{nopt} \in \{1, 10\}$ (since these 2 performed best in preliminary experiments), and the dim of the mixing network in $\{64, 256\}$. We set $\lambda_{nopt} = 10$ for *5m_vs_6m* and $\lambda{nopt} = 1$ for *3s5z*. The dim of the mixing network is set to 64.

For MASAC we consider $\alpha_{ent} \in \{0, 0.001, 0.01\}$ and set it to 0 for *3s5z* and 0.01 for *5m_vs_6m*.

## C.3 SMAC Super Hard Maps

We consider $\alpha \in \{0.01, 0.1, 0.5, 0.75\}$ and set $\alpha = 0.5$ for OW-QMIX and $\alpha = 0.75$ for CW-QMIX.

For the experiment in Figure 6 we only considered $\alpha \in \{0.5, 0.75\}$ and set $\alpha = 0.75$ for both methods.

# D  Ablations

In order to better understand our method, we examine 3 additional baselines:

**QMIX + $\hat{Q}^*$.** This ablation removes the weighting in our loss ($w = 1$), but still uses $\hat{Q}^*$ to bootstrap. This ablation allows us to see if a better bootstrap estimate alone can explain the performance of WQMIX.

Figure 7: Median test return for the predator prey environment, comparing Weighted QMIX and 3 ablations.

Figure 8: Median test win % with an increased rate of exploration, comparing Weighted QMIX and 3 ablations.

**QMIX + CW/OW.** This ablation introduces the CW/OW weightings into QMIX's loss function for $Q_{tot}$. We do not additionally learn $\hat{Q}^*$.

# E Results

In this section we present the results of additional experiments that did not fit in the main paper.

**Ablation experiments for Predator Prey.**

Our ablation experiments in Figure 7 show the necessity for both $\hat{Q}^*$ and a weighting in order to solve this task. As expected QMIX $+\hat{Q}^*$ is unable to solve this task due to the challenges of relative overgeneralisation. The use of a uniform weighting in the projection prevents the learning of an optimal policy in which two agents coordinate to capture a prey. Thus, even if $\hat{Q}^*$ can theoretically represent the $Q$-values of this optimal policy, the QMIX agents are unable to recover it. Figure 7 also shows that QMIX with just a weighting in its projection (and no $\hat{Q}^*$) is unable to successfully solve the task.

**Ablation experiments testing robustness to increased exploration.**

Figure 8 shows the results of further ablation experiments, confirming the need for both $\hat{Q}^*$ and a weighting to ensure consistent performance. Note in particular that combining QMIX with a weighting results in significantly worse performance in *5m_vs_6m*, and no better performance in *3s5z*.

**Effect of $\alpha$ on performance.**

Figure 9 shows the effect of varying $\alpha$ in the weighting function for CW-QMIX and OW-QMIX. We can see that if $\alpha$ is too low, performance degrades considerably.

Figure 9: Median test win % with an increased rate of exploration. **Above:** The effect of varying $\alpha$ for CW-QMIX. **Below:** The effect for OW-QMIX.