[Reviews · NeurIPS 2020]

Review 1

Summary and Contributions: This paper extends QMIX to use weights when projecting the joint actions into the mixing network. They first show QMIX minimizes the squared error across all actions, and that this causes the learning to fail to find the optimal policy in general even with access to the true state-action value function. To address this, they introduce a weighting into this projection to emphasize better joint actions with two schemes: Centrally-Weighted QMIX and Optimistically-weighted QMIX. These are both proven to recover the optimal maximum-value joint action, while also requiring learning a true joint state-action value function Q* along with the mixing network.

Strengths: I liked this paper, it showed an inherent flaw in QMIX and provides a scalable approach to address it. While on the surface it appeared as a simple extension, the analysis and guarantees make it a solid contribution in my opinion. All of the claims were sound (but I didn't check the proofs in supplementary), and it improved SOTA for SMAC. This is highly relevant to the MARL community.

Weaknesses: Why not include QTRAN, MADDPG, or MASAC on any of the SMAC experiments? Especially QTRAN as it seemed to be competitive on predator-prey (well within the quartile of OW-QMIX).

Correctness: I found no issues

Clarity: Yes, the paper is well written and easy to understand. The plots are very small, they would be impossible to view if printed.

Relation to Prior Work: Mostly, but I think a short related work section should be included (maybe combined with Background section for space?). It is clearly related to QMIX (which it extends) and VDN (very related factorization algorithm). I see there's an extended related work section in the supplementary material, but I still think you could summarize in the paper, at least to contrast with QTRAN.

Reproducibility: Yes

Additional Feedback: 222: How does using a QMIX approximation to Q* work? I guess you could try multiple best joint actions from Q_tot to evaluate in Q*? And won't this cause the policy induced by Q* to be different? It's very interesting to see QMIX performing best on the corridor scenario of SMAC (fig 4), and that the weighted QMIX can perform nearly as well with a modified architecture. I see in the supplementary material the modified architecture uses a hypernetwork layer—does this mean it is restricted to postive weights like QMIX? 137: Use MSBE without defining it


Review 2

Summary and Contributions: This paper aims to address the limitations of QMIX, that is unable to represent joint action-value functions that are characterized as nonmonotonic. To overcome this limitation, this paper proposes the QMIX operator to mix Q targets together as Q_tot using two kinds of weighting functions. The modified version of QMIX outperforms vanilla QMIX and other MARL methods in two test domains.

Strengths: The author uses a tabular example of QMIX to show its limitations and then draws their proposed weighted QMIX. The two kinds of weighting functions are theoretically analyzed to show Q_tot recovers the optimal policy. Weighted QMIX outperforms vanilla QMIX and other MARL methods in two test domains.

Weaknesses: The weighted QMIX only modifies QMIX by using a weighting function to get the Q_tot, and the two kinds of weighting functions seem too simple, so the contribution seems incremental. Extra computation cost also restricts its scalability.

Correctness: Yes

Clarity: Yes

Relation to Prior Work: Yes, but the related work seems missing. And some related work should be also compared or at least discussed. [1] Q-value Path Decomposition for Deep Multiagent Reinforcement Learning. ICML. 2020. [2] Qatten: A General Framework for Cooperative Multiagent Reinforcement Learning. arXiv preprint arXiv:2002.03939.

Reproducibility: Yes

Additional Feedback: AFTER REBUTTAL After reading the authors' response and fellows' reviews. I keep my score unchanged. The authors show the poor performance of weighted QMIX on some SMAC scenarios and describe that it is due to the extra complexity. So the extra experiments may be incompatible with this rebuttal's explanation. I am not convinced about the results given in the rebuttal. Why weighted QMIX performs much better on bane_vs_bane than on 5m_vs_6m as bane_vs_bane seems a harder scenario.


Review 3

Summary and Contributions: In this paper, the authors formalize the objective that QMIX optimizes and introduce the QMIX operator to abstract the algorithm procedure. Based on the formalization, they present several limitations of QMIX under idealized circumstances and introduce Weighted QMIX operator, which is proved theoretically correct to improve the performance. The realization of Weighted QMIX for deep RL is later tested in different scenarios and its improvement compared to other algorithms is demonstrated in the end.

Strengths: 1. Theoretically, the paper formalizes the optimization objective in QMIX and states its limitation in detail. Then it further proposes the method using weighted squared error and unrestricted target function to fix the representational limitation in QMIX which can only project the Q_{tot} as a monotonic function. The theoretical proof they provided also gives the construction of the parameter \alpha. 2. In experiments, the article has ablations which provide more specific understanding in the use of different components in the architecture. 3. On the relevance to the NeurIPS, this article is about Dec-POMDP and the MARL method and therefore is related to NeurIPS.

Weaknesses: 1. The proof of your theory lacks discussion of POMDP settings. Although the framework in focused in solving the Dec-POMDP problem, most parts of the proof are under MDP setting. 2. In the ablation part, the performance of QMIX+\hat{Q}^{*} has a significance difference between two environments in Figure 8. But there is no more discussion on that phenomenon. The use of weighting is not that convinced. 3. In Section 6.2.3, the performance of the Weighted QMIX method is unacceptable. The authors argue that the complexity introduced by \hat{Q}^* is responsible for the regression in performance. However, they don’t analyze the difference between the corridor task and other above-mentioned tasks, and they fail to present how the difference was related to the limitation of \hat{Q}^*. 4. About the parameter α of weighing Function, although the author gives a basis for selection in the appendix, the value of α seems not to be verified. And If the selection of α in the appendix is based on reliable evidence, you might be able to introduce it briefly when you introduce the weighting function. 5. The description of the experiment is a little brief. Some readers without the relevant knowledge will be confused when they read the section. 6. Some formulas in this paper are not standard. For example, the first condition in the formula 4 may need to be modified.

Correctness: This paper has sufficient theoretical analysis and proof, combined with the author's experimental results. I think it is correct.

Clarity: Overall, the paper is well written and organized. I list some confusing details below: 1. In Abstract, the agent’s ordering of actions is pointed to be important in representing value functions. But the proposed architecture seems to be incapable for dealing that case. 2. As you mentioned from line 117 to line 119, when using the original QMIX algorithm, sampling uniformly from a replay buffer does not strictly lead to a uniform weighting schema. However in the realization of Weighted QMIX you provided in Section 5, the loss in Equation 8 also suffers from the same problem. 3. In line 165, the exploration of every state-action pair is the same thing which is done in WQMIX. 4. In line 535, u^{*} is not defined before. 5. In line 86, R should be r. 6. In line 609, why do you use \hat{Q}^* instead of Q^* here? 7. In Figure 3, why is the variance so big?

Relation to Prior Work: The relationship is quite clear. WQMIX is an improved version of QMIX. To be specific, the difference between this work and the previous work is as follows: 1. The mix part of the target network is no longer subject to monotonicity constraints. 2. The loss function is calculated by adding weights to each state-action pair.

Reproducibility: No

Additional Feedback: 1. In Section 6, why do not you use QTRAN, MADDPG and MASAC to do some experiments? 2. How will input s to the Mix network portion be handled during execution? 3. How does the content of lines 163-167 relate to context?


Review 4

Summary and Contributions: This paper aims to solve the shortage in QMIX that the monotonic mixing of each agent’s utilities prevents QMIX from representing value functions in which an agent’s ordering over its actions can depend on other agents. The main contribution in this paper is the weighted QMIX algorithm which learns a weight of each agent in the QMIX framework, purposing two weighting central weighting and optimistic weighting and achieving good performance. Furthermore, this paper has proved the weighting formulation can reach the true optimal actions. Although the time for reviewing manuscripts is very tight, I think the whole work is good and interesting although there are several problems which confusing me.

Strengths: The main contribution in this paper is the weighted QMIX algorithm which learns a weight of each agent in the QMIX framework, purposing two weighting central weighting and optimistic weighting and achieving good performance. Furthermore, this paper has proved the weighting formulation can reach the true optimal actions.

Weaknesses: 1. The Joint Q function (Critic) seems to be learned from a feed-forward network from local Q function, but in the QMIX, there is a mixing network. Is this different between Weighted QMIX and QMIX ? Can you prove that the formulation of directly using nn to calculate the critic is better than mixing network ? 2. Can you explain more on the relationship between your work between the work of “QOPT: Optimistic Value Function Decentralization for Cooperative Multi-Agent Reinforcement Learning”. This work also consumes about the weight in value based MARL framework like QMIX etc. Furthermore, in that work, they use a weighted mixing network. And in their experiment, the critic part directly using nn to estimate the global optimal Q value will have a weak performance in some tasks.

Correctness: yes

Clarity: yes

Relation to Prior Work: Can you explain more on the relationship between your work between the work of “QOPT: Optimistic Value Function Decentralization for Cooperative Multi-Agent Reinforcement Learning”. This work also consumes about the weight in value based MARL framework like QMIX etc. Furthermore, in that work, they use a weighted mixing network. And in their experiment, the critic part directly using nn to estimate the global optimal Q value will have a weak performance in some tasks.

Reproducibility: Yes

Additional Feedback:

[Author Response · NeurIPS 2020]

We thank the reviewers for their feedback on our sub-
mission. We have fixed a config error when running OW-
QMIX on Predator Prey (there is now very little vari-
ance). We have also run additional experiments demon-
strating significantly better performance of weighted
QMIX on another *hard* SMAC map *bane_vs_bane*.
**Reviewer 1:** >*"Why not include QTRAN, MADDPG,*
*or MASAC on any of the SMAC experiments?"*
We have already included QTRAN, MADDPG and
MASAC on the SMAC experiments in Figure 2. Due to

(Left) Corrected experimental results for OW-QMIX on
Predator Prey. (Right) Weighted QMIX vs QMIX on
`bane_vs_bane` for $\epsilon$ annealed over 50k and 1mil.

their relatively poor performance there we did not run them on the *Super-Hard* SMAC maps (Figures 3 and 4) due to
the large computational cost of those experiments.
>*"How does using a QMIX approximation to $Q^*$ work?...And won't this cause the policy induced by $Q^*$ to be different?"*
In order to enable tractable maximisation, we use our QMIX approximation ($Q_{tot}$) to $Q^*$ to suggest the best joint action.
In general, the greedy action for $Q^*$ and $Q_{tot}$ can differ during training. However, Theorems 1 and 2 prove that given
sufficient training (and an appropriate $\alpha$) they will be the same.
>*"...the modified architecture uses a hypernetwork layer—does this mean it is restricted to positive weights like QMIX?"*
Yes, the first layer of $\hat{Q}^*$'s mixing network is restricted to non-negative weights, but $\hat{Q}^*$ is not restricted to being
monotonic due to subsequent layers.
**Reviewer 2:** >*"The weighted QMIX only modifies QMIX by using a weighting function to get the $Q_{tot}$, and the two*
*kinds of weighting functions seem too simple, so the contribution seems incremental."*
We disagree that the simplicity of the weighting function makes our approach too incremental. The use of a weighting
function in order to train a monotonic approximation to a learned unrestricted $Q^*$ is a significant algorithmic change
over QMIX. Additionally, we have proven that the two weighting functions we have considered are guaranteed to ensure
the maximal joint action is correct (given sufficient training and an appropriate $\alpha$) in contrast to QMIX which can fail to
recover the optimal joint action for the simple matrix game in Table 2. Furthermore, the framework we have introduced
for analysing Weighted QMIX can be used to analyse QTRAN and explain its empirical performance.
>*"Extra computation cost also restricts its scalability."*
Compared to QMIX, during training we must perform inference and train an additional model (with the same complexity
as QMIX). This does **not** restrict the scalability of Weighted QMIX compared to QMIX, as demonstrated by our
experiments on `bane_vs_bane` featuring 24 agents. We will include a discussion of the two papers you have provided.
**Reviewer 3:** >*"The proof of your theory lacks discussion of POMDP settings."* We deliberately restricted our
theoretical analysis to the MMDP setting in order to avoid the additional complexity of partial observability. The
MMDP setting allows for a cleaner presentation that focuses on our main goal of analysing the effect of the limited
representation of QMIX on the learned $Q_{tot}$ (and thus the learned policy).
>*"...performance of QMIX+$\hat{Q}^*$ has a significance difference ... in Figure 8... The use of weighting is not that convinced."*
On `3s5z` the weighting does not affect performance, but on `5m_vs_6m` it has a significant effect and on Predator Prey
every method without the weighting is unable to solve the task, showing that it is crucial to our method.
>*"In Section 6.2.3, the performance of the Weighted QMIX method is unacceptable."* That's the point: Section 6.2.3
aims to show the limitations of our method, which we believe is important for identifying areas for future research.
>*"The authors argue that the complexity introduced by $\hat{Q}^*$ is responsible for the regression in performance."*
Figures 4 and 5 demonstrate a clear performance difference for Weighted QMIX when **only** changing the architecture
used to represent $\hat{Q}^*$. This provides evidence that the poor performance is due to the architecture used to represent $\hat{Q}^*$.
>*"...$\alpha$ of weighing Function, although the author gives a basis for selection in the appendix, the value of $\alpha$ seems not to*
*be verified."* Our theoretical results only show that there exists an $\alpha$ which works in **all** cases. They are not intended to
provide a method for selecting an appropriate $\alpha$. We will discuss the selection of $\alpha$ for experiments in the main paper.
>*"... agent's ordering of actions is pointed to be important in representing value functions. But the proposed architecture*
*seems to be incapable for dealing that case."* $\hat{Q}^*$ is capable of representing **any** joint action $Q$-value function.
>*"... sampling uniformly from a replay buffer does not strictly lead to a uniform weighting schema. However in the*
*realization of Weighted QMIX you provided in Section 5, the loss in Equation 8 also suffers from the same problem."*
A uniform weighting is an assumption we make to simplify our analysis (to make it clearer). It is not required for the
Deep RL realisation of QMIX or Weighted QMIX.
>*"How does the content of lines 163-167 relate to context?"* Lines 163-167 explain why the failure modes discussed in
Section 3.1 are problematic. In particular, they show fundamental limitations of QMIX that cannot be addressed without
a significant algorithmic change, even in the idealised setting we consider. We will fix points 4, 5, and 6 on Clarity.
>*"How will input s to the Mix network portion be handled during execution?"*
During decentralised execution only the agent parts of $Q_{tot}$ are required. We do not need access to the state, the mixing
network, or $\hat{Q}^*$ during execution.

[Meta-Review · NeurIPS 2020]

I want to thank the authors for preparing the detailed rebuttal. This paper was discussed among all the reviewers during the post-rebuttal discussion phase. Also, given the borderline scores, we requested an additional emergency reviewer for this paper. While the rebuttal helped clarify some of the reviewers' questions, the reviewers shared a few concerns regarding the experimental evaluation, comparisons to SOTA, and the relationship of the proposed approach w.r.t. the relevant literature. Overall, the reviewers have a positive assessment of the paper and appreciated the technical insights to design the weighted QMIX algorithm. Based on these discussions, the reviewers have updated their reviews, providing additional feedback --- I hope the authors will find this useful. I would like to strongly encourage the authors to incorporate the reviewers' feedback when preparing the paper's final revision.